# Association between radiographic severity with health-related quality of life in elderly women with knee osteoarthritis: A cross-sectional study

Jiulong Song[1☉], Ziqi Ye[2☉], Wen Li[3], Zihao Chen[1], Xinwei Wang [4,*], Wei Chen[5*]

**1** College of Physical Education, Yangzhou University, Yangzhou, Jiangsu, China, **2** Department of Rehabilitation Medicine, Nanjing First Hospital, Nanjing Medical University, Nanjing, Jiangsu, China, **3** Nanjing Zijin Hospital, Nanjing, Jiangsu, China, **4** School of Sports Science, Nanjing Normal University, Nanjing, Jiangsu, China, **5** First Department of Acupuncture and Moxibustion, The Affiliated Hospital of Jiangxi University of Chinese Medicine, Nanchang, Jiangxi, China

☉ These authors contributed equally to this work
* 904233232@qq.com (WC); 18851729275@163.com (XW)

## Abstract

### Background and aims

Knee osteoarthritis (OA) is a common chronic condition among the elderly, leading to a decline in OA patients' quality of life. This study aimed to investigate the relationship between radiographic severity and health-related quality of life (HRQoL) in elderly women with knee OA.

### Methods

A total of 80 elderly women with knee OA were enrolled in this study. Radiographic severity was assessed with the Kellgren-Lawrence (K/L) scale, we divided the subjects into early (1–2) and late (3–4) according to the K/L stage. HRQoL assessment was conducted using the MOS item Short-Form 36 (SF-36). The association of HRQoL with knee OA severity was estimated using logistic regression. Applied a random forest model to assess the importance and accuracy of relevant variables in the occurrence of OA. The LASSO (Least Absolute Shrinkage and Selection Operator) regression was then used to identify key factors associated with OA, which were incorporated into the development of a risk prediction nomogram model. Furthermore, a receiver operating characteristic (ROC) curve was constructed to evaluate the model's discriminative ability for OA.

### Result

The mean age of the patients was 64.7±6.74 years, and the mean course of disease was 5.01±2.12 years. HRQoL score (SF-36 PCS and MCS) was significantly worse in the late-stage group compared to the early group ($p < 0.05$). The late group K/L

**Data availability statement:** The original contributions presented in the study are included in the article. The data supporting the findings of this study are available in Dryad. DOI: https://doi.org/10.5061/dryad.sj3tx96ff.

**Funding:** The author(s) received no specific funding for this work.

**Competing interests:** The authors have declared that no competing interests exist.

scale has a negative correlation with SF-36 PCS (r = -0.598) and MCS (r = -0.625) and a strong positive correlation. In logistic regression analysis, the K/L scale were significantly associated with SF-36MCS (OR = 0.86, *p* = 0.041), SF-36 PCS (OR = 0.85, *p* = 0.025) and TUG (OR = 1.80, *p* = 0.001). The nomogram model based on key OA risk factors identified by LASSO regression demonstrated substantial predictive value for OA, with an area under the curve (AUC) of 72.2%.

## Conclusion

The radiographic severity of knee OA was correlated with health-related quality of life. The HRQoL is an important predictive indicator of the severity of knee OA severity, which might provide beneficial management and treatment for patients with knee OA.

---

## 1. Introduction

Knee osteoarthritis (OA) is one of the most prevalent chronic joint diseases among the elderly and a leading cause of functional impairment and disability [1,2]. Epidemiological studies show that 8.1% of older adults in China experience symptomatic knee OA [3], and the prevalence of knee OA in females (10.3%) was higher than that in males (5.7%) [4,5]. As the global population continues to age, knee OA has emerged as a significant public health problems [6,7]. The development of knee OA is influenced by cumulative exposure to various risk factors, such as age, obesity, trauma, and biological changes in joint structure [8]. Pathological features of knee OA include the degeneration of articular cartilage, osteophyte formation, and synovium inflammation, all of which contribute result in joint dysfunction [9]. Common clinical symptoms of knee OA include pain, stiffness, and limitation of physical activity [10], all of which contribute to diminished physical function and quality of life [11].

At present, radiography is the most widely used method for diagnosing knee OA. Standardized grading systems, such as the Kellgren-Lawrence (K/L) scale, enables clinicians to assess the degree of joint damage based on radiographic features, including cartilage narrowing, osteophyte formation, and subchondral sclerosis [12]. Although radiographs provide valuable insights into structural changes in the joint, they fail to capture the clinical manifestations of the knee OA [13]. Many patients with knee OA show minimal or no symptoms, despite radiographic findings may be consistent with a clinical diagnosis of OA.

Health-related quality of life (HRQoL) in patients with knee OA is frequently compromised due to pain, reduced mobility, and the psychological burden associated with chronic illness [14]. A recent study compared the relevant quality of life of OA patients with healthy individuals [15]. The results indicated that OA patients scored lower in physical health, physical activity, and mental health compared to healthy controls [16,17]. Furthermore, HRQoL was an important factor in the assessment and management of knee OA, particularly in the late stages, as it plays a crucial role in determining the need for joint replacement.

However, some studies have reported that the association between the radiographic severity of knee OA and clinical outcomes remains controversial [18–21]. Additionally, there is a lack of evidence regarding the relationship between radiographic features and HRQoL in patients with knee OA. Therefore, this study aims to investigate the association between radiographic severity and HRQoL in elderly women with knee OA. Based on these findings, a nomogram model was developed to assess its potential value in diagnosing of knee OA.

## 2. Method

### 2.1. Study design

This was a single-center, prospective cross-sectional study. Between July 21, 2021 to June 18, 2023, 80 patients with knee OA were recruited from the Maigaqiao Community Health Service Center in Qixia District, Nanjing, China. The study was approved by the Nanjing Sport Institute Committee (KYCX20–1571), and all participants provided written informed consent to participate in this study.

### 2.2. Participants

Inclusion criteria: an elderly woman older than 50 years, with a clinical radiological diagnosis of knee OA [22], and knee pain for at least 6 months. Exclusion criteria: patients who had undergone hip, knee, and ankle joint surgery within the past 6 months, received knee joint injections or medication in the last 3 months, had impaired consciousness, or were unable to cooperate with the study procedures.

### 2.3. Assessments

**Radiography diagnosis:** Anteroposterior and lateral radiographs of both knees were obtained for each participant. The radiographic severity was graded according to the Kellgren-Lawrence(K/L) scale by two professional radiologists with 3 and 6 years of diagnostic. The K/L scale is a widely accepted and reliable grading system for knee OA, endorsed by the World Health Organization [23]. The K/L scale ranges from 0 to 4, where 0 = normal radiograph; 1 = joint space narrowing, suspected osteophyte; 2 = joint narrowing, obvious osteophyte; 3 = moderate joint narrowing, clear osteophyte; 4 = severe joint space stenosis and formation of a large number of osteophytes. K/L scores of 1 or 2 were classified as early-stage OA, and scores of 3 or 4 were classified as late-stage OA [24].

**Body Mass Index (BMI):** According to the international uniform classification standards recommended by the World Health Organization [25], participants were classified into the following 5 grades: Normal (18.5–23.9 kg/m2); Overweight (24–29.9 kg/m$^2$); Mildly obese (30–34.9 kg/m$^2$); Moderately obese (35–39.9 kg/m$^2$); and Severely obese (≧40 kg/m$^2$).

**West Ontario and McMaster University Osteoarthritis Index (WOMAC):** The WOMAC scale is a validated measure of OA severity [26]; it includes pain, stiffness, and physical function assessment [27]. The scale included 5 pain items (0–50), 2 stiffness items (0–20), and 17 physical function items (0–170). The reliability of the WOMAC pain and physical function subscales in this study was high (ICC = 0.86 and 0.89, respectively) [26].

**Timed Up and Go (TUG) test:** The TUG is a fast and quantitative measure of physical function. Participants sit in a chair about 45 cm high. Upon receiving the start of the instructions, they stand up from the chair and walk at their normal walking speed of 3 meters. Walk around the sign, turn back to the chair, and then turn around and sit down [28].

**Health-related quality of life(HRQoL) assessment:** HRQoL was evaluated using the MOS item Short-Form 36 (SF-36). The SF-36 is currently used widely to assess the general population's quality of life and evaluate clinical trial effects. It includes eight domains and the internal consistency of each subscale was excellent (Cronbach's coefficient alpha range 0.64 to 0.93). Our study used the Physical Components Summary (PCS) and Mental Components Summary (MCS) as evaluation indicators [29].

**Self-Rating Anxiety Scale (SAS) and Self-Rating Depression Scale (SAS):** The SAS and SDS were used to assess symptoms of anxiety and depression. Each scale includes 20 items, with a 1–4 scale; a higher score indicates more severe symptoms. Cronbach's alpha for the SAS and SDS in this study was 0.83 and 0.73, respectively [30,31].

## 2.4. Sample size

The sample size for this study was estimated using G-power software (Version 3.1). A two-sided t-test was chosen to compare the difference between two independent groups. The calculation was based on an assumed Effect Size of 0.71, a significance level a of 0.05, and a statistical power of 0.8. Taking into account a potential 20% attrition rate, the required sample size was calculated to be at least 40 participants per group, resulting in a total sample size of 80 participants.

## 2.5. Statistical analysis

Descriptive statistics were used to summarize the demographic characteristics of the participants. Categorical variables were expressed as counts and percentages, while continuous variables were summarized as means and standard deviations. The normality of continuous data was assessed using the Shapiro-Wilk test. For normally distributed data, differences between the early (K/L grade 1 or 2) and late (K/L grade 3 or 4) groups were assessed using the independent t-test. For non-normally distributed data, the Mann-Whitney U-test was applied. Additionally, Analysis of Covariance (ANCOVA) was used to adjust for potential confounding factors. Spearman's correlation analysis was applied to assess the relationships between the radiographic K/L grade and continuous variables. The strength of correlations was interpreted according to Cohen [32], $r = 0$ indicates no correlation, $r = 0.20$ to 0.50 indicates a weak to moderate correlation, and $r = 0.50$ to 0.80 indicates a strong correlation. Logistic regression analysis was employed to determine whether variables such as age, BMI, WOMAC, physical function, and quality of life could predict the severity of knee OA. The random forest model with 500 trees and 3 variables per split was used to evaluate the importance and predictive accuracy of relevant variables in the occurrence of knee OA. Additionally, the Least Absolute Shrinkage and Selection Operator (LASSO) regression was performed to identify key predictors and reduce dimensionality, thus mitigating multicollinearity. Model evaluation and parameter selection were performed using 10-fold cross-validation, with the optimal regularization parameter (λ) determined by minimizing model deviance. Based on the significant variables identified by LASSO regression, a logistic regression model was developed, and a nomogram was created to visually represent the contribution of each variable to predicting knee OA severity. The predictive performance of the nomogram was validated using the Receiver Operating Characteristic (ROC) curve and the Area Under the Curve (AUC). An AUC value greater than 0.7 was considered indicative of good predictive performance. All statistical analyses were performed by using SPSS 25.0(IBM SPSS Inc., Chicago, USA) and R software (version 4.4.1 R Foundation; Vienna, Austria). The setting test level of the study α=0.05, $p < 0.05$ that the difference was statistically significant.

## 3. Result

A total of 80 female patients, with a mean age of 64.7±6.74 years (range, 51–80 years) and an average course of disease of 5.01±2.12 years (range, 1–12 years), were enrolled in this study. According to the radiograph K/L grading system, 6 patients (7.5%) had grade 1; 41 patients (51.25%) had grade 2; 26 patients (32.5%) had grade 3; and 7 patients (8.75%) had grade 4. Demographic data, BMI scores, WOMAC scale scores, TUG, SF-36 PCS, SF-36 MCS, SAS, and SDS details are shown in Table 1.

When participants were grouped according to the K/L scale into early (1–2) and late (3–4) stages, there was no significant difference in age and course of the disease between the two groups. Compared with the early group, the late group had significantly higher BMI ($p = 0.002$), WOMAC pain ($p = 0.001$), stiffness ($p = 0.001$), physical function($p = 0.010$), and TUG scores($p = 0.003$). Additionally, the late group had a lower quality of life score (SF-36 PCS and MCS) ($p < 0.05$) and higher anxiety (SDS) score and depression (SAS) score than the early group ($p < 0.05$) (Table 2).

**Table 1. Demographic characteristics of the patients.**

| Variable | Total (n = 80) | Min-max |
|---|---|---|
| Age (years, mean ± SD) | 64.70 ± 6.74 | 51-80 |
| Course of disease (years) | 5.01 ± 2.12 | 1-12 |
| Hight (cm, mean ± SD) | 160.30 ± 5.39 | 147-176 |
| Weight (kg, mean ± SD) | 63.70 ± 10.16 | 38-105 |
| BMI (kg/m$^2$,) | 24.74 ± 3.62 | 17.1-43.7 |
| Radiograph score, n (%) | | |
| K/L grade 1 | 6 (7.5) | |
| K/L grade 2 | 41 (51.25) | |
| K/L grade 3 | 26 (32.5) | |
| K/L grade 4 | 7 (8.75) | |
| WOMAC (mean ± SD) | | |
| Pain (0–50) | 18.53 ± 5.62 | 10-34 |
| Stiffness (0–20) | 8.92 ± 3.68 | 2-19 |
| Physical Functions (0–100) | 37.65 ± 8.96 | 16-58 |
| TUG (s, mean ± SD ) | 10.43 ± 1.12 | 8.16-14.13 |
| SF-36 PCS (0–100) | 66.34 ± 9.78 | 32.75-87.50 |
| SF-36 MCS (0–100) | 70.67 ± 12.22 | 31.0-94 |
| SAS (20–80) | 38.43 ± 5.03 | 28-53 |
| SDS (20–80) | 40.53 ± 7.58 | 26-54 |

SD: standard deviation, n: number of patients; BMI: Body Mass Index; K/L: Kellgren-Lawrence scale; WOMAC: The Western Ontario and McMaster Universities Osteoarthritis Index; TUG: Time up and Go Test, SF-36: The MOS item short form health survey; MCS: Mental component summary; PCS: Physical component summary; SAS: Self-rating Anxiety Scale; SDS: Self-rating Depression Scale.

**Table 2. Comparison of demographic and clinical characteristic, WOMAC, TUG SF-36, SAS and SDS of OA patients between early group (K/L 1-2) and late group (K/L 3-4) radiological stages.**

| Variable | Early group (K/L1-2) | Late group (K/L3-4) | P value |
|---|---|---|---|
| Age (years) | 64.33 ± 6.63 | 65.46 ± 6.68 | 0.454 |
| Course of disease, years | 4.76 ± 2.32 | 5.15 ± 1.87 | 0.455 |
| BMI (kg/m$^2$) | 23.88 ± 2.75 | 26.53 ± 3.84 | 0.002 |
| WOMAC | | | |
| Pain (0–50) † | 17.07 ± 3.86 | 21.57 ± 6.29 | 0.001 |
| Stiffness (0–20) † | 7.90 ± 3.01 | 10.36 ± 4.10 | 0.001 |
| Physical Functions (0–100) † | 35.57 ± 8.51 | 40.73 ± 8.56 | 0.010 |
| TUG (s) | 10.16 ± 0.76 | 10.98 ± 1.27 | 0.003 |
| SF-36 PCS (0–100) ‡ | 68.66 ± 9.97 | 61.52 ± 7.38 | 0.002 |
| SF-36 MCS (0–100) ‡ | 72.67 ± 6.77 | 66.52 ± 7.35 | 0.025 |
| SAS (20–80) † | 37.40 ± 4.19 | 40.57 ± 5.55 | 0.008 |
| SDS (20–80) † | 38.62 ± 7.28 | 44.51 ± 5.97 | 0.001 |

Values are shown as mean ± SD (standard deviation); BMI: Body Mass Index; K/L: Kellgren-Lawrence scale; WOMAC: The Western Ontario and Mc-Master Universities Osteoarthritis Index; TUG: Time up and Go Test; SF-36: The MOS item short form health survey; MCS: Mental component summary; PCS: Physical component summary; SAS: Self-rating Anxiety Scale; SDS: Self-rating Depression Scale. † Lower scores indicated a lighter state. ‡ Higher scores indicated an improved state.

Table 3 presents the associations between age, course of disease, BMI, WOMAC, TUG, SF-36, SAS, and SDS scores in the early and late-stage OA groups. No significant associations were found in the early-stage group with regard to age, disease duration, stiffness, and SF-36MCS score in the early group of patients with knee OA. The late group has a negative correlation with SF-36 PCS (r = -0.598) and MCS (r = -0.625) and a strong positive correlation with age (r = 0.456), course of the disease (r = 0.636), BMI(r = 0.451), pain(r = 0.423), stiffness(r = 0.630), physical function (r = 0.535), TUG(r = 0.626), SAS(r = 0.570) and SDS(r = 0.551).

The several independent variables (age, BMI, WOMAC, TUG, and quality of life) having significant correlations with the severity of knee OA were examined in a regression model to determine their relative influence on the severity of knee OA. Logistic regression analysis revealed the severity of knee osteoarthritis was significantly associated with age (OR = 1.14, $p = 0.025$), BMI (OR = 1.45, $p = 0.002$), WOMAC pain (OR = 1.29, $p = 0.007$), SF-36MCS (OR = 0.86, $p = 0.041$), SF-36 PCS (OR = 0.85, $p = 0.025$) and TUG (OR = 1.80, $p = 0.001$). No associations were observed between the severity of knee osteoarthritis with WOMAC stiffness, physical function, and quality of life. See Table 4 for details.

To assess the importance and accuracy of 10 variables in predicting the occurrence of OA, a random forest model was employed. Among these, the SF-36 MCS demonstrated the highest importance and predictive accuracy (Fig 1). Subsequently, we developed a risk prediction model based on LASSO penalized regression, incorporating all covariates to identify parameters most strongly associated with OA (Fig 2). LASSO regression achieves feature selection by introducing an L1 regularization term (absolute value penalty) into the ordinary least squares regression, causing some coefficients to shrink toward zero, thereby isolating the most significant features or variables. The coefficient shrinkage in LASSO regression is achieved by minimizing the loss function combined with the L1 regularization term, which encourages certain coefficients to reduce to zero, effectively excluding the corresponding features (Fig 2A). The cross-validation curve was used to select the optimal lambda value, ultimately retaining 8 variables, which enhanced the model's stability and improved its predictive ability (Fig 2B). Based on the eight variables(age, BMI, physical function, TUG, SF-36 MCS, SF-36PCS, SDS and SAS) most closely associated with OA identified through LASSO regression, we constructed an OA risk prediction nomogram model, which incorporates HRQoL variables, demonstrated a high predictive capability (Fig 3A). Its predictive performance for OA risk model was validated through the ROC curve (AUC = 72.2%) (Fig 3B).

**Table 3. Correlation between early group (K/L 1-2) and late group (K/L 3-4) radiological stages with age, course of disease, BMI, WOMAC, TUG, SF-36, SAS and SDS.**

| Variable (n = 80) | Early group (K/L1-2) | Late group (K/L3-4) |
|---|---|---|
| Age (years, mean ± SD) | 0.137 | 0.456* |
| Course of disease, years | 0.090 | 0.636** |
| BMI | 0.413* | 0.451** |
| WOMAC | | |
| Pain | 0.298* | 0.423* |
| Stiffness | 0.220 | 0.630** |
| Physical Functions | 0.319* | 0.535** |
| TUG | 0.411** | 0.626** |
| SF-36 PCS | -0.309* | -0.598** |
| SF-36 MCS | -0.182 | -0.625** |
| SAS | 0.310* | 0.570** |
| SDS | 0.426** | 0.551** |

*: $p < 0.05$; **: $p < 0.01$; BMI: Body Mass Index; K/L: Kellgren-Lawrence scale; WOMAC: The Western Ontario and McMaster Universities Osteoarthritis Index; TUG: Time up and Go Test; SF-36: The MOS item short form health survey; MCS: Mental component summary; PCS: Physical component summary; SAS: Self-rating Anxiety Scale; SDS: Self-rating Depression Scale.

**Table 4. Results of logistic regressions of K/L score with Age, BMI, WOMAC, physical function and quality of life.**

| Variable (n = 80) | OR | 95%CI | P value |
|---|---|---|---|
| Age | 1.14 | 1.01 to 1.22 | 0.025* |
| BMI | 1.45 | 1.14 to 1.84 | 0.002** |
| WOMAC Pain | 1.29 | 1.07 to 1.55 | 0.007** |
| WOMAC Stiffness | 1.18 | 0.90 to 1.53 | 0.211 |
| WOMAC Physical Functions | 1.03 | 0.94 to 1.12 | 0.524 |
| TUG | 1.80 | 1.52 to 2.16 | 0.001** |
| SF-36 PCS | 0.86 | 0.81 to 1.02 | 0.041* |
| SF-36 MCS | 0.85 | 0.83 to 1.03 | 0.025* |
| SAS | 1.12 | 0.96 to 1.32 | 0.083 |
| SDS | 1.08 | 0.96 to 1.20 | 0.085 |

*: $p < 0.05$, **: $p < 0.01$, BMI: Body Mass Index, WOMAC: The Western Ontario and McMaster Universities Osteoarthritis Index, TUG: Time up and Go Test; SF-36: The MOS item short form health survey; MCS: Mental component summary; PCS: Physical component summary; SAS: Self-rating Anxiety Scale; SDS: Self-rating Depression Scale; OR: Odds ratio; 95%CI: 95% confidence intervals.

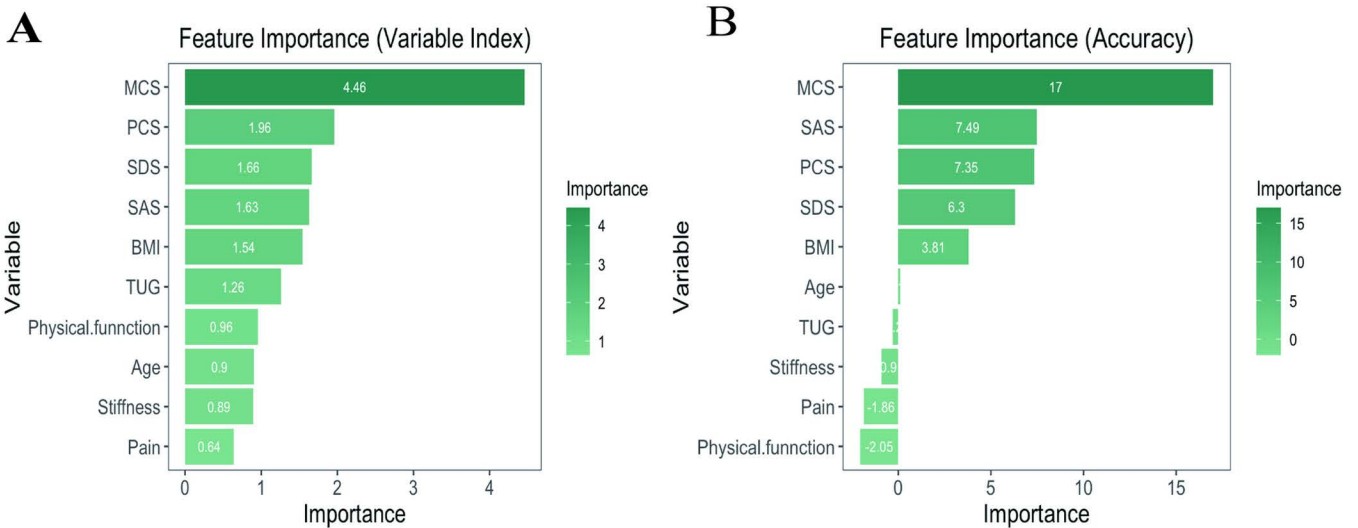

**Fig 1. Random forest plot.** (A) The importance of the 10 variables associated with the occurrence of OA risk. (B) The accuracy of the 10 variables associated with the occurrence of OA risk.

## 4. Discussion

This study explored the relationship between radiographic severity and health-related quality of life in elderly women with knee OA. The results indicated that the late-stage group exhibited significantly higher BMI, pain, stiffness, and TUG scores, along with lower quality of life scores and higher levels of anxiety and depression compared to the early-stage group. In addition, increasing radiographic severity was negatively correlation with quality of life and positively correlated with age, course of the disease, BMI, pain, stiffness, physical function, TUG, SAS, and SDS. A random forest model was employed to assess the importance and predictive accuracy of 10 variables for OA occurrence. LASSO regression identified eight key factors strongly associated with OA risk. Based on these factors, a nomogram model was developed, which demonstrated good performance in predicting the risk of OA.

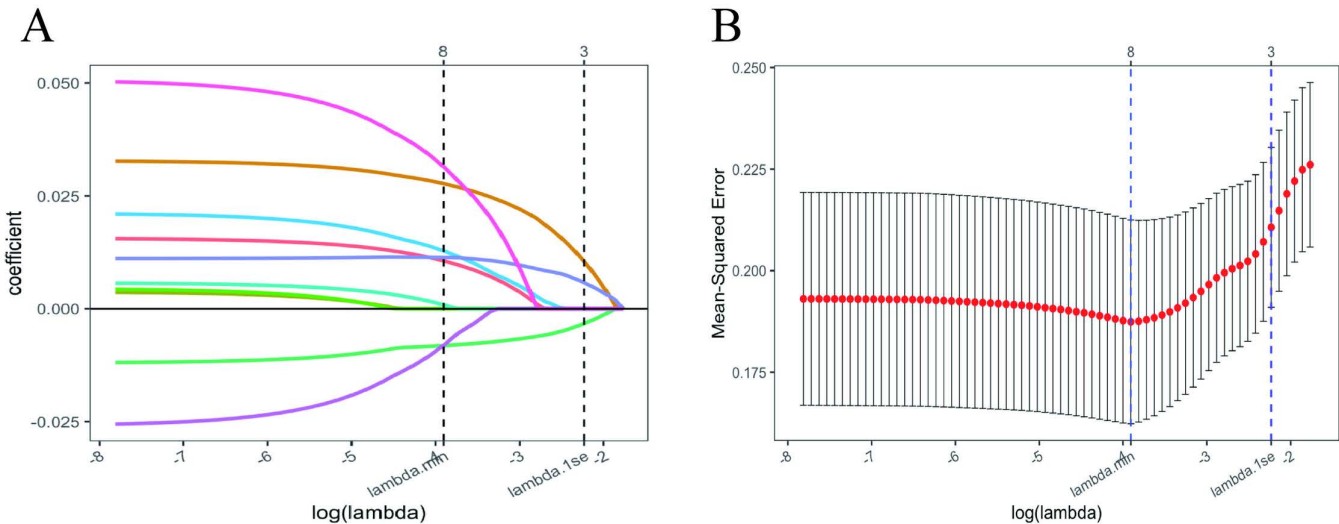

**Fig 2. The LASSO regression analysis used to identify factors associated with OA.** (A) The coefficient shrinkage process for all relevant variables, with changes in coefficients represented by lines of different colors, indicating the variations of each feature at different levels of shrinkage. (B) Ten-fold cross-validation of the LASSO regression model. LASSO: Least Absolute Shrinkage and Selection Operator.

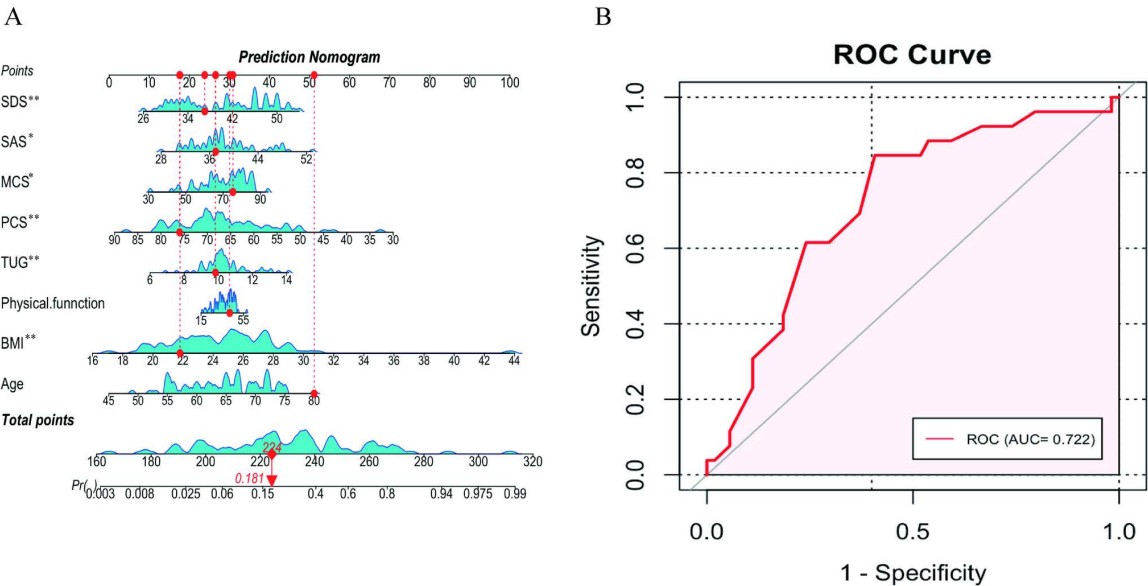

**Fig 3. The establishment and validation of the OA risk prediction model.** (A) A nomogram model representing the eight factors associated with OA risk. (B) The ROC curve used to assess the nomogram model's predictive ability for OA risk. *$P<0.05$, **$P<0.01$.

The K/L grading system is a crucial indicator for assessing the severity of knee OA, primarily reflects pathological changes such as cartilage loss, osteophyte formation, and joint space narrowing [33]. Previous studies have shown that a higher BMI in patients with knee OA is often associated with more severe radiographic findings, including higher K/L grades [34]. Our study found that the late-stage group had a significantly higher BMI, which was strongly positively correlated with radiographic severity. This suggests that an elevated BMI may contribute to the worsening radiographic

appearance of knee OA. Clinicians should, therefore, recommend weight management as an important strategy for the treatment of OA to slow the progression of the disease. A study has also indicated that individuals with a high BMI have a significantly higher risk of knee OA compared to those with a normal BMI [35]. This is consistent with our observation of a strong correlation between BMI and the radiographic severity of knee OA. Felson et al. [36] found that BMI levels were significantly higher in patients with symptomatic knee OA than in those with asymptomatic knee OA. Recent studies have demonstrated that higher BMI is an independent predictor of knee OA risk [37].

In addition, this study found that TUG time increased with the severity of the radiographic disease, suggesting that as radiographic severity worsens, the patients' functional status gradually declines. The TUG test can be utilized clinically as a routine tool for assessing functional decline, thereby facilitating the planning of appropriate intervention measures. Hirotaka et al. [38] also reported a positive correlation between TUG time and K/L grade in knee OA patients, confirming that the TUG test is a reliable indicator of functional status of knee OA patients. A prospective study of knee OA patients demonstrated that prolonged TUG times were not only associated with joint structural changes but also strongly correlated with self-reported levels of functional impairment, pain severity, and decline in quality of life [39]. Thus, the TUG test reflects not only the deterioration of muscle strength and mobility in knee OA patients but also serves as a sensitive early indicator of functional decline.

HRQoL is a crucial indicator for assessing the overall health status of patients with chronic diseases [40]. Studies have shown that, compared to the radiographic features of knee OA, HRQoL better reflects patients' clinical symptoms and the extent of their functional limitations [41]. Our findings further suggest that, unlike OA grading based solely on imaging, HRQoL assessment provides more comprehensive information on patients' functional status and psychological burden. Our results indicate that HRQoL is significantly associated with the knee OA severity, as evidenced by a negative correlation between the SF-36 PCS and MCS scores and disease progression. These findings suggest that HRQoL assessment can be used clinically to predict the OA severity, enabling early identification of high-risk patients and the implementation of more targeted treatment strategies. A previous study identified significant differences in quality of life scores among knee OA patients at various stages, with those in advanced stages showing markedly lower HRQoL scores compared to those in the early stages [42]. Our findings also support this observation. The reduced quality of life in knee OA patients has been shown to be closely associated with pain, functional limitations, and impaired social relationships. The effects of pain and disability may lead to anxiety and depression, which further contributing to the decline in quality of life [43,44].

Health quality of life is influenced not only by physical health conditions but also by various factors such as social support and mental health [45]. Anxiety and depression are commonly observed in knee OA patients and may exacerbate pain and mobility limitations, thus negatively impacting their quality of life [46]. Recent studies suggest that psychological states not only affect pain perception but may also worsen disease symptoms through inflammatory responses and immune mechanisms [47]. Our study found that a significant positively correlation between anxiety and depression levels and OA severity. Research has shown that depressive symptoms in knee OA patients are closely linked to pain, functional limitations, and a decreased quality of life [48]. Anxiety and depression not only affect the patient's emotional and physiological states but may also worsen knee joint inflammation by altering immune responses and the neuroendocrine system, thereby accelerating the progression of OA [49]. Therefore, in clinical practice, doctors should not only focus on the physical condition of patients with OA, but also pay close attention to their mental health issues.

This study also applied advanced machine learning models. Using the random forest model, we revealed that the SF-36 MCS exhibited the highest importance in predicting the occurrence of OA, indicating that HRQoL factors can serve as a sensitive predictive indicator of knee OA severity. This finding is consistent with the study by Yasmin et al. [50], which also identified HRQoL as a major predictive factor for the onset and increased burden of knee OA. Furthermore, the LASSO regression model selected eight variables most closely associated with OA, including age, BMI, physical function, TUG, SF-36 MCS, SF-36 PCS, SDS, and SAS. The nomogram model based on LASSO regression further confirmed this, with HRQoL variables being the primary contributors to the model's predictive capability (AUC = 72.2%). This finding

indicates that HRQoL assessment can not only predict the severity of knee OA but also help identify high-risk patients who may require psychological intervention or social support, thereby optimizing treatment strategies and offering strong support for clinical decision-making.

Although this study provides important insights into the relationship between knee OA severity and quality of life, there are still several limitations. First, while a sample size of 80 patients is suitable for preliminary analysis, the relatively small cohort may limit the generalizability of the results. Future studies should employ a multicenter, large-sample design to further validate our findings. Second, as this study used a cross-sectional design, it is unable to establish causal relationships between the associated factors and the severity of knee OA. Subsequent research could explore the dynamic changes in these factors and their causal relationships with disease progression. Finally, our study focused on an elderly female population, which may influence the analysis of gender factors in patients with knee OA. Additionally, the results of this study cannot be generalized to the male gender or to individuals under the age of 50. Future studies should include more diverse populations to enhance the applicability of the findings.

## 5. Conclusion

The results of this study confirmed a significant correlation between radiographic severity and health-related quality of life in elderly women with knee OA. The application of LASSO regression analysis and the development of a nomogram model helped identify the HRQoL factors most closely associated with OA occurrence. By integrating various critical OA-related factors, the model demonstrated the ability to accurately predict OA risk. Therefore, incorporating HRQoL assessment alongside clinical and radiographic findings may provide valuable insights for the management and treatment of patients with knee OA.

## Supporting information

**S1 File. Data.**
(XLSX)

**S2 File. Early Group data.**
(XLSX)

**S3 File. Late group data.**
(XLSX)

## Acknowledgments

We would like to thank all participants in the Maigaoqiao Community Health Service Center in Qixia District, Nanjing City, China.

## Author contributions

**Data curation:** Ziqi Ye.

**Investigation:** Zihao Chen.

**Resources:** Wen Li, Wei Chen.

**Software:** Jiulong Song, Zihao Chen.

**Supervision:** Ziqi Ye, Wen Li, Wei Chen.

**Validation:** Xinwei Wang.

**Writing – original draft:** Jiulong Song.

**Writing – review & editing:** Xinwei Wang.

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
