## [Decision Letter · Decision Letter 0]

12 Jan 2025

PONE-D-24-57302Association between radiographic severity with health-related quality of life in elderly women with knee osteoarthritis: A cross-sectional studyPLOS ONE

Dear Dr. Wang,

Thank you for submitting your manuscript to PLOS ONE. After careful consideration, we feel that it has merit but does not fully meet PLOS ONE’s publication criteria as it currently stands. Therefore, we invite you to submit a revised version of the manuscript that addresses the points raised during the review process.

We look forward to receiving your revised manuscript.

Kind regards,

Fatih Özden, PT, PhD, Assoc. Prof,

Academic Editor

PLOS ONE

6. Please include your tables as part of your main manuscript and remove the individual files. Please note that supplementary tables (should remain/ be uploaded) as separate "supporting information" files

Additional Editor Comments:

The reviewers of the paper suggested minor revisions. I suggest you to submit the revised paper regarding their suggestions. Best Wishes

Reviewers' comments:

Reviewer's Responses to Questions

**Comments to the Author**

1. Is the manuscript technically sound, and do the data support the conclusions?

Reviewer #1: Yes

Reviewer #2: Yes

2. Has the statistical analysis been performed appropriately and rigorously? 

Reviewer #1: Yes

Reviewer #2: Yes

3. Have the authors made all data underlying the findings in their manuscript fully available?

Reviewer #1: Yes

Reviewer #2: Yes

4. Is the manuscript presented in an intelligible fashion and written in standard English?

Reviewer #1: Yes

Reviewer #2: Yes

5. Review Comments to the Author

Reviewer #1: Dear authors, I would like to congratulate you on this valuable manuscript. However, I have some concerns.

1) In the sentence “However, some studies have reported that the association between the radiographic severity of knee OA and clinical outcomes remains controversial[18].”, you have used plural expression by writing “some studies” although you have only one reference. Please review this again.

2) Improve your manuscript English. Use subjects more specifically in sentences instead of words like they, their, it. If necessary, you can utilize the translation service.

3) Indicate which statistical analysis you utilized to determine whether there is a normal distribution.

4) In the limitations section, state that the results of the present study cannot be generalized to the male gender and to individuals younger than 50 years of age.

5) In your findings, the body mass index is significantly higher in the group with KL 3-4 than in the other group. This may have negatively affected quality of life and other outcome measures, regardless of the stage of CL. How did you control for the possible negative effect of body mass index in the comparison between groups?

Best regards,

Reviewer #2: I have provided my two comments below.

1-The authors should provide a calculation-based sample size analysis.

2-Clinical implications section should be added to the discussions section to improve the practical implications of the results.

6. PLOS authors have the option to publish the peer review history of their article (what does this mean? ). If published, this will include your full peer review and any attached files.

**Do you want your identity to be public for this peer review?** For information about this choice, including consent withdrawal, please see our Privacy Policy .

Reviewer #1: No

Reviewer #2: No

---

## [Author Response · Author response to Decision Letter 1]

20 Jan 2025

Dear Editor and Reviewers,

We would like to express our sincere gratitude for the time and effort editor and the reviewers have spent on evaluating our manuscript titled " Association between radiographic severity with health-related quality of life in elderly women with knee osteoarthritis: A cross-sectional study". We truly appreciate the constructive feedback and thoughtful suggestions provided, which have significantly improved the quality of our work.

In response to the editor and reviewers' comments, we have carefully revised the manuscript to address the concerns raised. Below is a summary of the major revisions we made:

Thank you for the remainder from the editor. We have adjusted the manuscript according to the PLOS ONE style templates to ensure that all file naming complies with the journal's formatting requirements. The necessary revisions have been completed, and the updated files are attached.

Thank you for your feedback. We have verified the funding information and updated the "Funding Information" section to ensure that the provided grant numbers match the actual funding.

According to PLOS ONE requirements, we have uploaded the minimal data set, which includes the raw data used to calculate means, standard deviations, and other statistical measures, as well as the values used to build graphs and the points extracted from images for analysis. We have deposited the data in the Dryad repository, and the relevant DOI is: http://datadryad.org/stash/share/zdrxjMWFkIjpRK4MB8XrQ9RzLDDO2gd6h0Uwh4pW5DA. The DOI link is: https://doi.org/10.5061/dryad.sj3tx96ff. Please feel free to contact us if you have any questions.

4. Your ethics statement should only appear in the Methods section of your manuscript. If your ethics statement is written in any section besides the Methods, please move it to the Methods section and delete it from any other section.

Thank you for the reminder. We have moved the ethics statement from other sections of the manuscript to the "Methods" section and ensured that the ethics statement appears only in the "Methods" section. Please refer to the resubmitted manuscript.

Thank you for your feedback. We have provided a separate caption for each figure to ensure that all figure descriptions are clear and accurate. The revised manuscript has been updated, please refer to it.

6. Please include your tables as part of your main manuscript and remove the individual files. Please note that supplementary tables (should remain/ be uploaded) as separate "supporting information" files

Thank you for your suggestion. We have uploaded all tables as part of the main manuscript and removed the separate table files.

7. Please review your reference list to ensure that it is complete and correct.

Thank you for your reminder. We have rechecked the reference list to ensure its accuracy and completeness.

Reviewer #1:

1) In the sentence “However, some studies have reported that the association between the radiographic severity of knee OA and clinical outcomes remains controversial [18].”, you have used plural expression by writing “some studies” although you have only one reference. Please review this again.

Thank you for your feedback. It was an oversight on our part, and we have now added additional relevant studies in this section to support the argument.

2) Improve your manuscript English. Use subjects more specifically in sentences instead of words like they, their, it. If necessary, you can utilize the translation service.

Thank you for your suggestion. We have carefully reviewed the manuscript and have avoided using ambiguous pronouns, ensuring that the subject of each sentence is clear and specific to improve the accuracy and readability of the language.

3) Indicate which statistical analysis you utilized to determine whether there is a normal distribution.

Thank you for your reminder. In the methods section, we have added that the Shapiro-Wilk test was used to assess the normality of the data, the specific method is described in the text.

4) In the limitations section, state that the results of the present study cannot be generalized to the male gender and to individuals younger than 50 years of age.

Thank you for your valuable suggestion. We have clearly stated in the limitations section that the study results are only applicable to specific age groups and genders, and therefore cannot be generalized to the male population or individuals under the age of 50. Future studies should include more diverse populations to enhance the applicability of the findings.

5) In your findings, the body mass index is significantly higher in the group with KL 3-4 than in the other group. This may have negatively affected quality of life and other outcome measures, regardless of the stage of CL. How did you control for the possible negative effect of body mass index in the comparison between groups?

Thank you for your question. We used Analysis of Covariance (ANCOVA) in the data analysis to control for the potential effect of BMI in the comparisons between groups. We have provided a detailed explanation of this in the methods section to ensure that the impact of BMI is appropriately controlled.

Reviewer #2:

1-The authors should provide a calculation-based sample size analysis.

Thank you for your suggestion. We have added detailed information about the sample size calculation in the methods section. Based on the expected effect size and statistical power analysis, we determined the required sample size to ensure sufficient statistical power for the study results.

2-Clinical implications section should be added to the discussions section to improve the practical implications of the results.

Thank you for your suggestion. We have added a section on clinical implications in the discussion, exploring the potential impact of the study’s findings on clinical practice and how they can guide future research.

---

## [Editor Report · Decision Letter 1]

4 Feb 2025

Association between radiographic severity with health-related quality of life in elderly women with knee osteoarthritis: A cross-sectional study

PONE-D-24-57302R1

Dear Dr. Wang,

We’re pleased to inform you that your manuscript has been judged scientifically suitable for publication and will be formally accepted for publication once it meets all outstanding technical requirements.

Kind regards,

Fatih Özden, PT, PhD, Assoc. Prof,

Academic Editor

PLOS ONE
---

## [Editor Report · Acceptance letter]

PONE-D-24-57302R1

PLOS ONE

Dear Dr. Wang,

I'm pleased to inform you that your manuscript has been deemed suitable for publication in PLOS ONE. Congratulations! Your manuscript is now being handed over to our production team.

Kind regards,

on behalf of

Assoc. Prof. Dr. Fatih Özden

Academic Editor

PLOS ONE